# The Influence of the Divergent Substrate on Physicochemical Properties and Metabolite Profiling of *Agrocybe cylindracea* Cultivation

**DOI:** 10.3390/jof11020132

**Published:** 2025-02-10

**Authors:** Hatungimana Mediatrice, Nsanzinshuti Aimable, Irambona Claude, Nyummah Fallah, Menna-Allah E. Abdelkader, Jules Biregeya, Yingping Hu, Lili Zhang, Hengyou Zhou, Jing Li, Penghu Liu, Zhanxi Lin, Dongmei Lin

**Affiliations:** 1National Engineering Center of Juncao Technology, College of Life Science, Fujian Agriculture and Forestry University, Fuzhou 350002, China; mediatunga@gmail.com (H.M.); nsanziaima@gmail.com (N.A.); claudebiofafu@gmail.com (I.C.); glo20081988@gmail.com (N.F.); menna.elsayed@alexu.edu.eg (M.-A.E.A.); biregeyakayihura2020@gmail.com (J.B.); hyp111419@126.com (Y.H.); lilizhang@fafu.edu (L.Z.); hengyou@fafu.edu.cn (H.Z.); fafulijing@fafu.edu.cn (J.L.); phliu1982@163.com (P.L.); lindongmei@fafu.edu.cn (D.L.); 2Rwanda Agriculture and Animal Resources Development Board, P.O. Box 5016 Kigali, Rwanda; 3Department of Genetics, Faculty of Alexandria University, Alexandria 21545, Egypt

**Keywords:** *Agrocybe cylindracea*, physicochemical properties, biological efficiency substrate

## Abstract

*Agrocybe cylindracea* is an important mushroom highly valued as a functional food for its nutritional and medicinal benefits. Many bioactive extracts from *A. cylindracea* have been found to exhibit antitumor and antioxidant activities. This research investigated the distinct substrates that affected the physicochemical and biocomponent properties and biological efficiency of *A. cylindracea*. The substrates used were 48% giant juncao grass mixed with 30% *Dicranopteris dichotoma* grasses, 48% wasted tea leaves mixed with 30% sawdust, and 78% cottonseed hulls, all supplemented with 20% wheat bran and 2% lime. The findings indicated that *A. cylindracea* grown in the cotton seed hulls had a significant biological efficiency, at 35.8%, followed by the GD + DD (31.5%) and WTL + SD (28.7%). The ash content peaked in the fruiting bodies grown on giant juncao grass and *D. dichotoma*, while the fat content was highest in those grown on giant juncao grass and *D. dichotoma*, followed by wasted tea leaves and sawdust. The protein content was significantly higher in the fruiting bodies cultivated on wasted tea leaves and sawdust, followed by cottonseed hulls. The carbon dioxide emissions varied across substrates, with the highest emissions observed during the maturity stage of the fruiting bodies grown on giant juncao grass with *D. dichotoma* and wasted tea leaves and sawdust. Emissions decreased sharply 110 days after cultivation. Essential metabolites, such as dopamine and caffeine, were enriched in the fruiting bodies grown on wasted tea leaves, sawdust, and cottonseed hulls. In contrast, tyramine and uracil were enriched in those grown on cottonseed hulls.

## 1. Introduction

Mushrooms are a macro fungus characterized by a prominent fruiting body that is large enough to be seen with the naked eye and can be harvested by hand. For centuries, mushrooms have been widely consumed as food and valued for their texture and various medicinal and tonic properties [1]. *Agrocybe cylindracea*, commonly referred to as the velvet foot mushroom, is an edible fungal species that has gained prominence in both culinary and medicinal contexts due to its substantial nutrient profile, distinctive flavor, and metabolites that have effects on tumors and chronic illnesses.

This species is also considered a versatile food supplement because of its abundant nutrients and bioactive compounds [2].

In the 21st century, the Asian continent has experienced a rapid increase in mushroom cultivation, with 9,854,391 tonnes per year, including *Agrocybe cylindracea*, particularly in China, which accounts for 90% of continental production [3], according to FAOSTAT released in 2021 [4]. Lichuan County in Jiangxi Province produces around 10,000 tons of fresh *Agrocybe cylindracea* mushrooms annually. These mushrooms are predominantly sold fresh and processed into dried products for distribution to major cities. China has identified *Agrocybe cylindracea* as a crucial species for agricultural industrialization projects, offering increased technical guidance and financial support to drive industrial development [5].

Mushrooms provide higher protein content than fruits and vegetables [6]. Growing mushrooms can significantly enhance sustainable livelihoods for the urban and rural poor [7], as they easily integrate with other income-generating activities while needing minimal physical and financial resources. Mushrooms are favored in human diets due to their good taste and their nutritive and healthy values. The mushroom fruit body is low in calories, fat, and cholesterol and rich in protein, carbohydrates, fibers, vitamins, and minerals. These nutritional properties make mushrooms a very beneficial dietary food [8]. *Agrocybe cylindracea* is an edible mushroom that belongs to the family of basidiomycetes and is in the order of Agaricales, with gills ranging from pink to dark brown [9]. Chen et al. explored the nutritional properties of *A. cylindracea* from different regions. They discovered many nutritional properties, including protein, dietary fiber, amino acids, fat, ash, and moisture [10]. Recently, ref. [11] extensively reviewed and reported the effects of substrate composition, strain, developmental stage, and ecological interactions of mycelial cells and fruiting bodies. These authors identified a wide variation in the concentrations of metabolites (e.g., ergosterol and phenolic compounds, etc.) in both the mycelial cells and fruiting bodies. A related investigation revealed that *A. cylindracea* fruiting bodies exhibit enhanced antioxidant activities, attributed mainly to ultra-high-pressure treatment and superfine grinding, indicating potential health benefits from their antioxidant properties [12].

The cultural materials known as substrates play a crucial role in determining the quality and yield of edible fungi. For successful mushroom cultivation, these substrates need to include cellulose, hemicelluloses, and lignins, as they provide essential sources of carbon, nitrogen, and other important minerals. This understanding of the substrate composition is fundamental for growers looking to optimize their mushroom production [13]. Generally, mushrooms can decompose organic matter, particularly cellulose and hemicellulose, and they produce a range of extracellular enzymes to facilitate this process. Therefore, different substrates can affect the fruiting-body growth of *A. cylindracea*, the development of biological efficiency, and the physicochemical and biocomponent proprieties [14]. Substrates, giant juncao grass [15], tea leaves [16], and cottonseed husks were used in this research to grow *A. cylindracea* [17]. Furthermore, there is a lack of comprehensive literature concerning the response of different growth stages of *A. cylindracea* to the response impacts on the physicochemical properties, biocomponents, yield, and biological efficiency. The research aimed to investigate how different substrates serve as growth materials for *Agrocybe cylindracea* cultivation and their impact on physiochemical properties and the metabolite profile of the fruiting bodies of *Agrocybe cylindracea*.

## 2. Material and Methods

### 2.1. Primary Spawn Preparation

The mushroom strain was obtained from the National Engineering of Juncao Center. To prepare the potato dextrose agar (PDA), 200 g of potatoes were peeled and chopped into small pieces. These pieces were boiled in 1000 mL of distilled water in a pressure cooker for approximately 15 min. The starchy liquid was then filtered and mixed with 20 g of dextrose and 20 g of agar. A total of 15 mL of the prepared PDA was dispensed into test tubes, which were subsequently sealed [18]. The solution was sterilized in an autoclave at 121 °C for 30 min and subsequently allowed to cool. A small fragment, approximately 10 mm, was extracted from a cultured strain of *A. cylindracea* for primary spawn inoculation. The inoculated test tubes were then incubated in a dark environment at a temperature of 25 °C.

### 2.2. Substrate Preparation and Cultivation

The substrates were sourced from the National Engineering of Juncao Center and were prepared as follows. The first substrate, designated as F1, consisted of 48% giant juncao grass (GJ) combined with 30% *Dicranopteris dichotoma* (DD). The second substrate, F2, contained 48% wasted tea leaves (WTL) mixed with 30% sawdust (SD). The third substrate, F3, comprised 78% cottonseed husks (CSH), with all substrates supplemented by 20% wheat bran and 2% lime, as demonstrated in Table 1. The components of the substrates were thoroughly mixed using a mixture machine, and sufficient water was added to achieve a moisture content of approximately 60% [19]. The polyethylene bags were then filled with between 1000 g and 1220 g of the prepared substrates. Each substrate was sterilized separately in an autoclave at 121 °C for 120 min and allowed to cool for 24 h. Following this, the inoculated polyethylene bags were transferred to a room maintained at 25 °C with 85% relative humidity for cultivation. The various substrates mentioned above were treated as distinct treatments. A total of 54 samples were generated at each time point.

Each replicate was homogenized and mixed to form a single composite sample. The fruiting bodies of *A. cylindracea* in each polyethylene bag were harvested at the maturity stage and weighed. We utilized the accumulated data to calculate the biological efficiency (BE), defined as the ratio of the weight of the fresh fruiting body of *A. cylindracea* (in grams) to the dry weight of the substrate (in grams), expressed as a percentage. The biological efficiency (BE) for each species was then calculated by the following formula [13].BE = Fresh weight of fruit body × 100Dry weight of the substrate

### 2.3. Determination of Enzyme Activity

The samples were collected from the initial developmental stages of the mycelium running stage through to the fruiting-body stage within the substrate, with three replicates. We sampled the inoculated bag at ten-day intervals to examine various enzyme activities during the growth and development phases of *A. cylindracea.* The determination of enzyme activity was conducted by using sinobestbio enzyme assay kits [20].

Four enzymes, namely carboxymethylcellulase, xylanase, laccase, and amylase, were determined during the different development stages of *A. cylindracea* in the different substrates. In the procedure for assessing carboxymethylcellulase activity, 1 g of the substrate mixed with the mycelium sample was measured, and 1 mL of the buffer solution was added and mixed. The mixture was then centrifuged at 8000 rpm for ten minutes at 4 °C, after which the supernatant was collected and placed on ice. The content of reducing sugars produced through the carboxymethyl cellulase-catalyzed degradation of cellulose was quantified using the 3,5-dinitrosalicylic acid method. The spectrophotometer was preheated for 30 min. And the wavelength was set to 550 nm, with distilled water used for calibration at zero [21]. Xylanase activity was evaluated by preparing D-xylose standard solutions (ranging from 100 to 600 µg/mL) and mixing 1 µL of D-xylose with 1.5 mL of DNS reagent. This mixture was boiled for 7 min and then cooled, after which the absorbance was measured at 550 nm [22].

Laccase activity was assessed by centrifuging 1 g of the substrate mixed with mycelium sample with an extraction solution at 10,000× *g* for 10 min at 4 °C. The control sample was prepared by combining 25 μL of boiled samples with 25 μL of test samples in a 2 mL EP tube, followed by the addition of 150 μL of working fluid containing ABTS to the test sample. An additional 150 μL of working fluid was added to the control, and both samples were incubated in a water bath at 60 °C for 3 min. The oxidation of ABTS was monitored by measuring the increase in absorbance at 420 nm and recorded every 30 s for a total duration of 180 s. The samples were analyzed using a spectrophotometer with a micro-quartz cuvette and a 96-well plate at room temperature [23]. To measure the total activity of alpha and beta amylase, a crude amylase solution was mixed with 4 mL of double-distilled water and thoroughly shaken. The released reducing sugars were quantified using a spectrophotometer/microplate reader, which was preheated for 40 min and the wavelength set at 540 nm [24].

### 2.4. Measurement of Lignocellulose Content and Carbon Dioxide Emission During A. cylindracea Cultivation

The Van Soest method was employed to quantitatively assess the lignocellulosic composition [25], specifically the contents of lignin, hemicellulose, and cellulose, utilizing 0.50 g of powdered substrate mushroom as the sample [26]. A square plexiglass cover with a total volume of 0.125 m^3^ was utilized for gas tracking. The cultivation box was sealed, and the control gas was monitored immediately after placing the glass cover. The test gas samples were collected one hour later in three parallel treatments by injecting gas into the prepared airbag. The gas was measured by using a gas chromatograph, from Agilent Technologies, R1778A machine [27]. The peak area and blank of three sets of parallel data were measured by gas chromatography (ppm value of carbon dioxide emissions)/(average value of peak area of carbon dioxide emissions). The peak area of the control was subtracted, and the resulting value was multiplied by the above data to give the ppm of carbon dioxide for the three sets of parallel data [28].

### 2.5. A. cylindracea Nutritional Composition Cultivated in Different Substrates

The nutritional composition of A. cylindracea was investigated following the method used by Dimopoulou et al. (2022) in their work. The contents of polysaccharides, fiber, carbohydrates, fat, proteins, amino acids, crude ash [29], and heavy metals (cadmium, arsenic, lead, and mercury) were performed in the fruit body of *A. cylindracea* [30]. A total of 0.05 g of the crushed *A. cylindracea* fruiting-body sample were combined with 1 mL of water in a test tube and homogenized in a water bath at 100 °C for 2 h. Following this, the mixture was centrifuged for 10 min at 10,000 rpm, and the supernatant was carefully removed. Subsequently, 0.2 mL of the supernatant were mixed with 0.8 mL of anhydrous ethanol. This resulting mixture was utilized to quantify the polysaccharide contents. The mixture is then measured at 490 nm using a spectrophotometer, as detailed in the method described by Zihao et al. (2021) in their work [17]. The determination of crude fiber was conducted following GB/T 5009.10-2003 [31] guidelines, while the crude ash content was determined based on GB/T12532-2008 [21,32]. The protein contents were determined as follows. The standard working solution was prepared by combining 100 volumes of BCA reagent A with 2 volumes of BCA reagent B. Reagent A was created by dissolving 1 g of sodium bicinchoninate, 2 g of sodium carbonate, 0.16 g of sodium tartrate, 0.4 g of NaOH, and 0.95 g of sodium bicarbonate in 50 mL of distilled water. The solution was brought to a final volume of 100 mL using distilled water, and the pH was adjusted to 11.25 by adding 10 M NaOH. Reagent B was prepared by dissolving 0.4 g of cupric sulfate (pentahydrate) in 5 mL of distilled water, which was then diluted to a total volume of 10 mL with additional distilled water. The absorbance of the known standard was measured at 562 nm. The fat content and amino acid composition of the fruiting body were analyzed using the RP-HPLC method, as outlined by Jia et al. (2020) in their work [24].

### 2.6. Screening of Metabolites/Bioactive Compounds During A. cylindracea Cultivation

The fruiting body of *A. cylindracea* was analyzed to determine its metabolite composition. For this analysis, 3 mg of the extract were combined with a solvent mixture of 90% dichloromethane (DCM) and 10% methanol, then homogenized. A Shimadzu GC-MS system was used for the analysis, featuring a capillary column (30 mm × 0.25 mm) and an electron ionization system set at 70 eV. Helium served as the carrier gas, flowing at a rate of 1 mL/min. An injection volume of 8.00 μL was used with a split ratio of 10:1, as described in the work of Tsiaka et al. [33]. The injector and ion source temperatures were 25 °C and 28 °C, respectively. The oven temperature began at 110 °C for 2 min, then ramped to 200 °C at 10 °C/min and to 280 °C at 5 °C/min, concluding with a 9 min hold at 280 °C. Mass spectra were scanned every 0.5 s, detecting fragments from 45 to 450 Da. The total run time was 36 min, calculating relative percentages based on peak areas, with interpretations using the NIST library. The sulfur content was measured using inductively coupled plasma tandem quadrupole mass spectrometry (ICP-QMS/QMS) [9].

## 3. Statistical Analysis

A principal coordinate analysis (PCoA) based on Bray–Curtis distances was performed to explore and visualize similarities and dissimilarities among metabolite communities. Later, we carried out a permutational multivariate analysis of variance (PERMANOVA) and paired the PERMANOVA with the “adonis” command in vegan at 999 permutations and α = 0.05 to understand the changes in the communities of metabolites. A volcano plot was conducted using an R language-based ggtern and grid to detect the enriched and depleted metabolites in the *A. cylindracea* subjected to various culture media. We performed a hierarchical clustering analysis using the “heatmap.2” function from the “gplots” R package. A bubble plot of the top 20 metabolites were statistically analyzed KEGG pathways was conducted using R software (R version 4.4.2). Pearson’s correlation coefficient was performed to test the association between the metabolites and the various essential elements/parameters identified during the growth and development of the *A. cylindracea* using R software. We performed a redundancy analysis (RDA) to evaluate the association between metabolite composition and various essential elements/parameters identified during the growth and development of the *A. cylindracea*. We used 999 permutations by employing the ‘vegan’ package to perform the significance difference. ANOVA was adopted to evaluate the test data. Later, we displayed the results using DPS software (version 7.05, www.dpssoftware.co.uk (accessed on 10 October 2024). Lastly, the changes between the mean values of each culture medium were explored using Tukey’s HSD test (*p* < 0.05) [34]. The graphs were designed and statistically analyzed using OriginPro (Version 2024)

## 4. Results

### 4.1. Enzyme Activities and Organic Compounds During A. cylindracea Cultivation

The enzyme activities of *A. cylindracea* at various growth stages, cultivated on different substrates, displayed distinct patterns. Figure 1A demonstrated that xylanase enzyme activity was significantly higher in the WTL + SD substrate at both 30 and 50 days post-cultivation. A similar trend was observed in the CSH substrate 60 days after cultivation. Most significantly, xylanase enzyme activity peaked during the primordial formation of *A. cylindracea* in the CSH substrate by reaching 1618.6 nmol/min/mg, substantially exceeding the levels observed in the WTL + SD and GJ + DD substrates.

Furthermore, the CMCase activity was significantly high (*p* < 0.05) in the GJ + DD substrate 10, 20, and 40 days after *A. cylindracea* cultivation. Similarly, CMCase activity demonstrated a similar trend in the WTL + SD 20, 30, and 50 days after *A. cylindracea* was cultivated. Moreover, CMCase activity exhibited a similar phenomenon in the CSH substrates at 30, 40, and 60 days of *A. cylindracea* cultivation compared with the WTL + SD and GJ + DD substrates. It is worth noting that CMCase activity significantly peaked (*p* < 0.05) (2642.5 μg/min/g) in the CSH substrates during the early stages of fruiting-body development (primordial formation), occurring 90 days after *A. cylindracea* cultivation. We noticed a similar pattern during the maturity stage of the fruiting body (120 days). Overall, CMCase activity decreased from *A. cylindracea* cultivation through its maturity stage (Figure 1B).

Figure 1C showed that GJ + DD exhibited the advantage of significantly increasing (*p* < 0.05) the laccase activity 10 d after *A. cylindracea* cultivation compared with CSH and WTL + SD treatments. Laccase activity in the CSH treatment marked a significant increase (*p* < 0.05) 20 and 60 d after *A. cylindracea* cultivation, followed by the early stages of fruiting-body development (primordial formation, i.e., 80 days after *A. cylindracea* cultivation compared with GJ + DD and WTL + SD. We also found that the laccase activity significantly increased (*p* < 0.05) in the WTL + SD substrate 30 and 60 days after *A. cylindracea* cultivation compared with the GJ + DD and CSH treatments. A similar pattern was noticed during primordia formation (i.e., 100 days after *A. cylindracea* cultivation) and the maturity stage (i.e., 120 days after *A. cylindracea* cultivation).

We also explored *A. cylindracea* amylase activity and found that it was significantly high (*p* < 0.05) in the WTL + SD treatment 30 and 50 days after *A. cylindracea* cultivation compared with the GJ + DD and CSH treatments. Amylase activity significantly peaked (*p* < 0.05) in the CSH treatment, especially during primordia formation, followed by the maturity stage of the *A. cylindracea*’s fruiting body (i.e., 120 d after *A. cylindracea* cultivation) (Figure 1D).

### 4.2. Organic Compound and Carbon Dioxide Emissions Were Detected During A. cylindracea Cultivation

The number of organic compounds was determined during the different growing stages of *A. cylindracea* subjected to the various substrates. The lignin and cellulose exhibited a descending pattern from the period of *A. cylindracea* cultivation to maturity in the different substrates. This phenomenon became more prevalent 20 d after *A. cylindracea* cultivation, with the primordial formation and the maturity stage of the fruiting body exhibiting the lowest lignin and cellulose. It is worth noting that this behavior was more pronounced in the GJ + DD compared with the WTL + SD and CSH treatments. Cellulose demonstrated a similar trend in the WTL + SD treatment, followed by the GJ + DD treatment (Figure 2A,B). Our results further showed that hemicellulose exhibited a descending trend from the period of *A. cylindracea* cultivation to the maturity stage. This phenomenon took 20 d after *A. cylindracea* cultivation, with the primordial formation and the maturity stage of the fruiting body exhibiting a significantly low amount of hemicellulose in the different substrates (Figure 2C). Figure 2D shows that the exhibited carbon dioxide emissions were slightly high when *A. cylindracea* was initially cultivated and decreased 20 d after cultivation. We also noticed that carbon dioxide emissions in the different substrates fluctuated, exhibiting an increasing trend, especially in the GD + DD and WTL + SD substrates. This pattern was more pronounced during the maturity stage of the fruiting body, followed by the primordia formation stage. However, carbon dioxide emissions sharply plummeted 110 d after *A. cylindracea* cultivation. Moreover, the total carbon content significantly diminished in the fruiting bodies subjected to the WTL + SD and CSH compared to the GD + DD substrates.

### 4.3. Biocomponents and Biological Efficiency of A. cylindracea Cultivated in the Different Substrates

We also observed that the protein content was significantly higher (*p* < 0.05) in the WTL + SD, followed by the CSH treatment, compared with the GD + DD, as is demonstrated in Figure 3A. It was also revealed that the dietary fiber and polysaccharide contents of the *A. cylindracea* fruiting bodies subjected to the WTL + SD exhibited no significant difference (Figure 3B,E). Figure 3C illustrates that the fat content in the *A. cylindracea* fruiting bodies subjected to the GD + DD substrates peaked, while the opposite was in the WTL + SD, followed by the CSH substrates. We also noticed that the ash content of the *A. cylindracea* fruiting body marked a significant increase in the GJ + DD compared with the WTL + SD and CSH substrates (Figure 3D). The nitrogen content significantly decreased in the *A. cylindracea*’s fruiting body grown in the WTL + SD compared with the GD + DD and CSH substrates. The total nitrogen revealed a similar trend in the *A. cylindracea* fruiting body subjected to the WTL + SD, followed by the CSH compared with the GD + DD. Further analysis revealed that the sulfur in the *A. cylindracea* fruiting body subjected to the WTL + SD culture medium marked a significant increase (*p* < 0.05) compared with the GD + DD and CSH substrates, as demonstrated in Figure 3F.

Figure 3G also demonstrated that the yield of the *A. cylindracea* subjected to the CSH treatment marked a significant increase (*p* < 0.05), followed by the WTL + SD relative to the GD + DD. We also investigated the biological efficiency (BE) during the entire growth period of the *A. cylindracea*. We observed that the *A. cylindracea* BE was significantly high (*p* < 0.05) in the CSH treatment, accounting for 35.8%, followed by the GD + DD (31.5%) and the WTL + SD (28.7%) culture medium (Figure 3I).

### 4.4. Metabolites Detected for A. cylindracea Grown in the Various Substrates

The performances of the metabolites (Appendix A) in each substrate relative to those in another treatment (Figure 3H,J,K, Appendix A) were evaluated. The analysis revealed that some essential metabolites, including dopamine and caffeine, were significantly enriched (*p* < 0.05) in the *A. cylindracea* subjected to the WTL + SD and CSH compared with the GJ + DD substrate. However, diacetyl and stearidonic acid depleted considerably in the *A. cylindracea* subjected to WTL + SD compared with the *A. cylindracea* subjected to the GJ + DD (Figure 3H,J, Appendix A). Moreover, tyramine and uracil were significantly enriched (*p* < 0.05) in the *A. cylindracea* subjected to the CSH compared with those subjected to the WTL + SD, while ergothioneine demonstrated the opposite (Figure 3K, Appendix A).

### 4.5. Composition and Abundance of Metabolites Detected During A. cylindracea Cultivation

The PCoA analysis was conducted to explore the metabolite compositions identified during the maturity stage of the *A. cylindracea* fruiting body subjected to the different substrates. We found that the distribution pattern of metabolite compositions was distinctly separated from each other, implying that the distribution pattern of these metabolite compositions was specific, as is demonstrated in Figure 4A. We also investigated the relative abundance of these metabolites and observed that glycerophosphocholine (25.94%), 16-hydroxy palmitic acid (16.47%), alpha-linolenic acid (7.59%), stearidonic acid (6.80%), dopamine (5.57%), L-Glutamine (5.29%), and caffeine (4.44%) were abundant. Others include l-isoleucine (4.38%), norepinephrine (3.83), acetylcarnitine (3.51%), D-Proline (3.35%), L-Pyroglutamic acid (1.89%), and L-Phenylalanine (1.84%) (Figure 4B).

A Venn diagram analysis further revealed that 14 (9.2%), 31 (20.3%), and 15 (9.8%) of the metabolites identified during the maturity stage of the *A. cylindracea* fruiting bodies were unique to the GJ + DD, WTL + SD, and CSH treatments, respectively. We also noticed that 20 (13.1%), 39 (25.5%), and 14 (9.2%) of the metabolites were common to both GJ + DD and WTL + SD, WTL + SD and CSH, and GJ + DD and CSH, respectively (Figure 4C).

Figure 5 illustrated that the metabolite composition was largely substrate-specific, which reinforced the pattern observed in Figure 4A. Figure 5 demonstrates that a significant number of these metabolites were detected during the maturity stage of the *A. cylindracea* fruiting bodies and were significantly expressed, especially those subjected to the WTL + SD and CSH compared with the GJ + DD treatment. For instance, N-Acetyl-D-Glucosamine 6-PhosphateL-Alanine, DL-lactate, Dihydroxyacetone phosphate, Uridine, L-Leucine, D-Arabinono-1.4-lactone, myoinositol, 4-Hydroxyphenylpyruvate, p-Hydroxyphenylacetic acid, Levulinic acid, Suberic acid, S-Methyl-5′-thioadenosine, (S)-2-Hydroxyglutarate, Nicotinamide adenine dinucleotide phosphate (NADP), L-Kynurenine, D-Threitol, Dihydroxyacetone, and D-Fructose, N2-Acetyl-L-ornithine were more pronounced in the *A. cylindracea* fruiting bodies under the WTL + SD compared with the CSH and GJ + DD treatments. We also observed that D-Ribulose 5-phosphate, Adenosine, Adenosine monophosphate (AMP), 2′-Deoxycytidine 5′-monophosphate (CMP), Uridine 5′-diphosphoqlucuronic acid (UDP.D-glucuronate), L-Phenylalanine, Hypoxanthine, Uracil, DL-3-Phenyllactic acid, Glycerol 3-phosphate, Uridine diphosphate glucose (UDP-D-Glucose), 3.4-Dihydroxyhydrocinnamic acid, L-Saccharopine, and D-Rose 5-phosphate demonstrated a similar trend in the *A. cylindracea* fruiting bodies subjected to the WTL + SD compared with the CSH and GJ + DD treatments. Moreover, Quinate, 3-Hydroxycapric acid, Dihydrothymine, L-Pyroglutamic acid, L-Glutamine, Allantoate/Allantoic acid, Uridine 5-monophosphate (UMP), L-Citrulline, L-Glutamate, Embelin, Cytidine, Guanosine, 3-Phosphoserine, Cytidine 5′monophosphate (CMP), Deoxycytidine, D-Lyxose, Adenine, Alpha-D-Glucose, D-Allose, and D-Mannose concentration were more significantly higher in the *A. cylindracea* fruiting bodies subjected to the CSH treatment compared with the WTL + SD and GJ + DD.

### 4.6. Metabolite-Enriched KEGG Pathways Identified During the A. cylindracea Cultivated in the Different Substrates

An enriched KEGG pathways analysis was performed further for the top 20 metabolites detected during the maturity stage of the *A. cylindracea* fruiting body. It was revealed that ABC transporters, pyrimidine metabolism, protein digestion and absorption, aminoacyl-RNA biosynthesis, mineral absorption, galactose metabolism, etc., were enriched significantly in the *A. cylindracea* fruiting bodies under the WTL + SD compared with the GJ + DD (Figure 6).

### 4.7. Metabolite Association with the Different Parameters Explored During A. cylindracea Maturity Stage

The association among some essential metabolites, enzyme activities, and vital *A. cylindracea* nutrients was established. It was noticed that metabolites, including caffeine, exhibited a significant positive association with xylanase, CMCase, amylase, laccase, and vital *A. cylindracea* nutrients, namely sulfur, nitrogen, and carbon. Correspondingly, stearidonic acid demonstrated a similar pattern with laccase and xylanase, followed by CMCase and amylase. Stearidonic acid positively correlated with sulfur, nitrogen, and carbon. Furthermore, sarcosine had a significant positive relationship with CMCase, amylase, and other nutrients, namely nitrogen and carbon. Nicotinate was significantly and positively associated with enzymes, namely xylanase, CMCase, and amylase, followed by nicotinate and vital *A. cylindracea* nutrients, including sulfur, nitrogen, and carbon dioxide. Glutathione exhibited a similar phenomenon with amylase, CMCase, and *A. cylindracea* nutrients, namely nitrogen and carbon (Figure 7A).

In addition, we tested the association of metabolites and *A. cylindracea*’s physicochemical properties, biocomponents, and productivity. The analysis demonstrated that tyramine exhibited a strong positive correlation with the *A. cylindracea* yield. However, polysaccharides and ash revealed the opposite. It was also found that dopamine had a significant positive correlation with the *A. cylindracea* yield. Caffeine was significantly and positively correlated with dietary fibers and proteins, whereas stearidonic acid exhibited the same trend with polysaccharides. In addition, ergothioneine was considerably and positively associated with proteins, followed by dietary fibers. On the other hand, ash demonstrated a contrary trend. Uracil was significantly and positively associated with the *A. cylindracea* yield, whereas uracil exhibited a strong negative relationship with ash and polysaccharides. Meanwhile, diacetyl demonstrated a similar pattern with dietary fibers. Additionally, sarcosine displayed a significant and positive association with the *A. cylindracea* yield and proteins. Nicotinate significantly and positively correlated dietary fibers and proteins. Glutathione exhibited a similar trend with yield and proteins, while ash revealed the opposite (Figure 7B).

We explored the relationships between metabolites and biochemical properties, as illustrated in Figure 8A, followed by the physicochemical properties in Figure 8B detected in the *A. cylindracea* fruiting bodies, to understand the role these different variables play in shaping metabolite compositions. It was revealed that metabolites, including L-Norleucine, ergothioneine, and L-Citrulline tended to favor dietary fibers and proteins, especially in the *A. cylindracea* fruiting bodies grown in the WTL + SD substrates. Further analysis using a permutation test analysis demonstrated that proteins had a significant positive correlation with these metabolites. A similar behavior was observed between xylitol and ash, as shown in Figure 8A. It was demonstrated that several metabolites, including ergothioneine, norleucine, and L-Citrulline tended to favor all the physicochemical properties detected in the *A. cylindracea* fruiting bodies, especially those subjected to the WTL + SD. A further analysis proved that carbon and nitrogen, followed by sulfur, xylanase, and carboxymethy cellulase exhibited a significant positive relationship with these metabolites, as indicated in Figure 8B.

## 5. Discussion

Growing evidence in scientific data has established that mushrooms’ growth, development, and productivity are responsive to various substrates. Adams et al. (2022) pointed out that different waste materials, including wheat bran and sawdust, significantly influenced *A. cylindracea* yield [35]. We found that the CSH culture medium had a significant *A. cylindracea* yield, followed by the WTL + SD growth medium, compared with the GJ + DD substrate. Combining sawdust with wasted tea leaves adds additional structural support and nutrients. Those findings were similar to the Ahmed et al. (2024) findings, which report that combining wasted tea leaves, sawdust, and cottonseed hulls with wood waste exhibited the advantage of boosting mushroom production, especially with a 3:1 ratio of wood waste to tea leaves. The resulting substrate provides an ideal environment for *A. cylindracea* propagation [36]. Several theories underpin this phenomenon: Sawdust’s, fibrous structure supports mushroom mycelium growth and development and facilitates the formation of fruiting bodies [37]. Moreover, the effects of the CSH provide a balanced substrate with structural integrity and nutrients.

Fruiting bodies of mushrooms have gained traction among many researchers, largely due to their nutritional and health benefits [38]. For instance, a study explored 11 species of fresh and dried, medicinal [39], and edible macrofungi and found that carbohydrates and proteins were more pronounced in the different species of mushrooms [40]. Chen et al. (2023) investigation also revealed that 43 *A. cylindracea* samples from 13 provinces in China exhibited differences in nutritional components of *A. cylindracea*. These authors documented that *A. cylindracea* had high amounts of protein and insoluble dietary fiber but low soluble dietary fiber and fat [2]. Our findings demonstrated that the ash content significantly peaked in the fruiting body of *A. cylindracea* that was subjected to the effects of the GJ + DD culture medium, while the fats of the *A. cylindracea* fruiting body exhibited the same behavior when subjected to the CSH, followed by the WTL + SD. Moreover, proteins in the *A. cylindracea* fruiting body cultivated in the WTL + SD, followed by the CSH, significantly increased. Those findings are parallel with Li et al.’s (2024) work, wherein the efficacy of the Korshinsk pea shrub on *A. aegerita* triggered a peak in polysaccharide and crude protein contents, by 4.46% and 26.60%, respectively, indicating an increase of 4.51% and 12.34% over the control. The marked increase in the nutritional composition of *A. cylindracea* subjected to the various effects of the different culture media could be ascribed to these substrate compositions and their nutrients composition [17].

Extensive investigations such as Claude et.al. (2024) have reported on the decisive role different substrates play (e.g., giant juncao grass and cottonseed hulls) in promoting the nutritional composition of mushrooms, which are closely similar to our findings [15].

Figure 2B showed that the carbon dioxide emissions were slightly high when *A. cylindracea* was initially inoculated into the various substrates and decreased 20 d after cultivation. We also noticed that carbon dioxide emissions in the substrate fluctuated, exhibiting an increasing trend, especially in the GD + DD and WTL + SD. This finding corroborates Li et al.’s work, revealing that spent mushroom substrate slightly increased the methane emissions but reduced the global warming potential of methane and nitrous oxide by 33.95% [41]. We believe that the ability of the mycelium of *A. cylindracea* to undergo cellular respiration, utilizing oxygen and releasing carbon dioxide as a product, contributed to this phenomenon, largely due to the mycelium’s ability to colonize the substrate and grow its metabolic activity, contributing to carbon dioxide emissions [42].

Studies have revealed that the bioactive active composition of living organisms, including plants [43] and eukaryotic organisms (mushrooms, yeasts, etc.), is responsive to substrate amendments [44]. Obed et al. (2022) found that substrates, especially sawdust and giant grass, influenced the bioactive components of oyster mushrooms, including flavonoids, saponins, triterpenoids, polyphenols, and steroids [45]. A volcano plot analysis revealed that the syngenetic effect of different substrates significantly enriched some essential metabolites [46]. The essential metabolites, including dopamine and caffeine, were significantly enriched in the *A. cylindracea* fruiting body subjected to the WTL + SD and CSH [47]. Moreover, tyramine and uracil were significantly enriched in the *A. cylindracea* fruiting body subjected to the CSH culture medium. The redundancy analysis (RDA) plots A and B show metabolomic data. L-Citrulline, Ergothioneine, and Glycerol compounds were most strongly associated with polysaccharides and fatty acids, having significant effects on the cellular metabolism of the *A. cylindracea* grown in the three different substrates [48].

This study deciphers the physicochemical and biocomponent composition of *A. cylindracea* at various growth stages when grown to different substrates. Notably, the analysis revealed that the highest biological efficiency (BE) was achieved using cottonseed husks (CSH), followed by giant juncao grass mixed with *D.Dichotoma.* Furthermore, the research illustrated dynamic changes in extracellular enzyme activity throughout mycelial development, primordia formation, and fruiting-body stages across different substrates, thereby enhancing the physicochemical properties of the mushrooms.

This study is highly relevant for researchers and mushroom cultivators aiming to utilize agricultural waste and giant grasses to boost productivity, improve physicochemical properties, and enhance the biocomponents of mushrooms, all while promoting environmental sustainability. Studies that focus on physiochemical and metabolite properties during various development stages in edible mushrooms are scarce. We recommend conducting further research on the changes in the physicochemical and metabolite properties of *Agrocyble cylindracea* at different developmental stages, examining the efficiency of carbon utilization in different substrate formulations, and exploring the influence of the substrate on flavor profiles.

## Figures and Tables

**Figure 1 jof-11-00132-f001:**
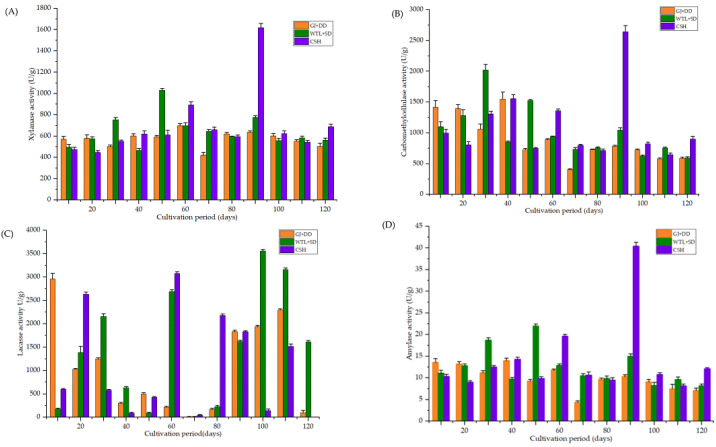
Enzyme activities (**A**–**D**) of *A. cylindracea* grown on giant juncao grass (GJ) combined with *Dicranopteris dichotoma* (DD), wasted tea leaves (WTL) combined (SD), and cottonseed hulls (CSH) supplemented with wheat bran and lime.

**Figure 2 jof-11-00132-f002:**
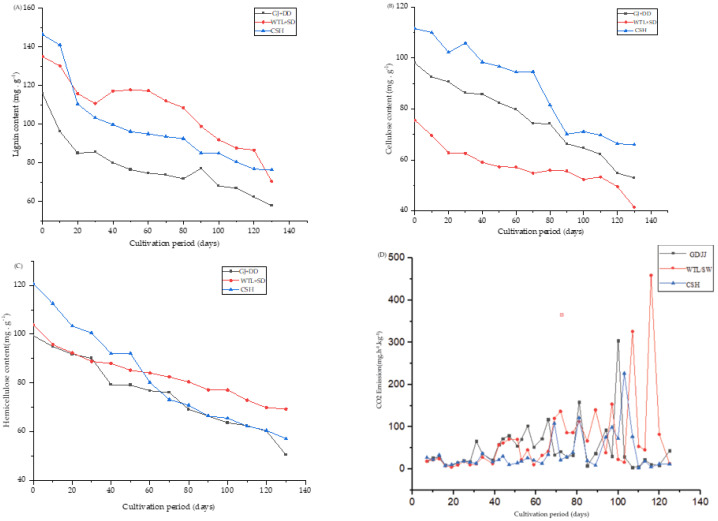
Organic compounds (**A**–**C**) and carbon dioxide emission (**D**) of *A. cylindracea* grown on giant Junco grass (GJ) combined with *Dicranopteris dichotoma* (DD), wasted tea leaves (WTL) combined (SD), and cottonseed hulls (CSH) supplemented with wheat bran and lime.

**Figure 3 jof-11-00132-f003:**
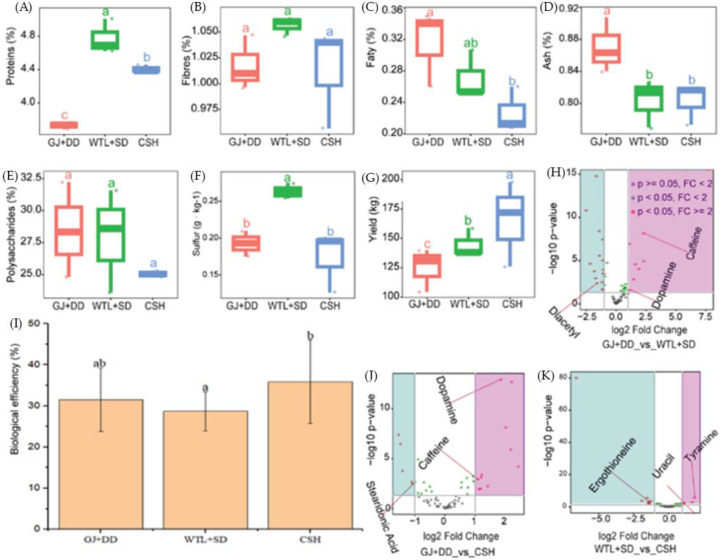
Essential elements (**A**–**F**), and productivity (**G**,**I**), *A. cylindracea* on various substrates. Different letters (a,b,c) indicate significant differences (*p* ≤ 0.05). Volcano plots showing the depleted (dark-greenish-blue color) and enriched (purple color) during *A. cylindracea* different growth stages detected in each substrate relative to those in another substrate (**H**,**J**,**K**).

**Figure 4 jof-11-00132-f004:**
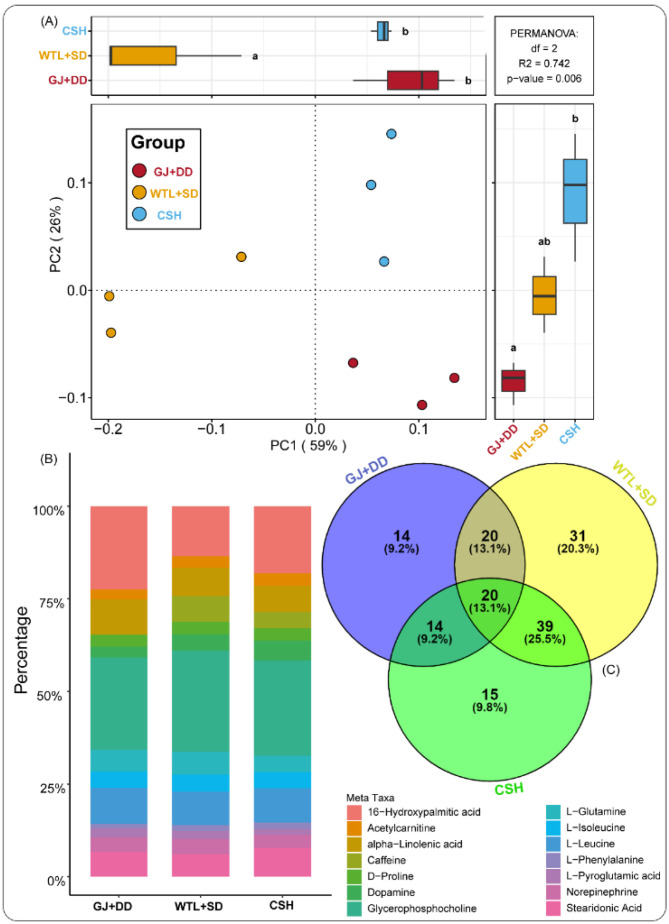
Principal component analysis (PCoA) illustrating metabolite composition detected during the maturity stage of *A. cylindracea* fruiting body subjected to the different substrates (**A**). Different letters (a,b) indicate significant differences (*p* ≤ 0.05). Bar graph revealing metabolites’ relative abundance during the maturity stage of *A. cylindracea* fruiting body under the various substrates (**B**). Venn diagram displaying the unique and overlapped metabolites (**C**).

**Figure 5 jof-11-00132-f005:**
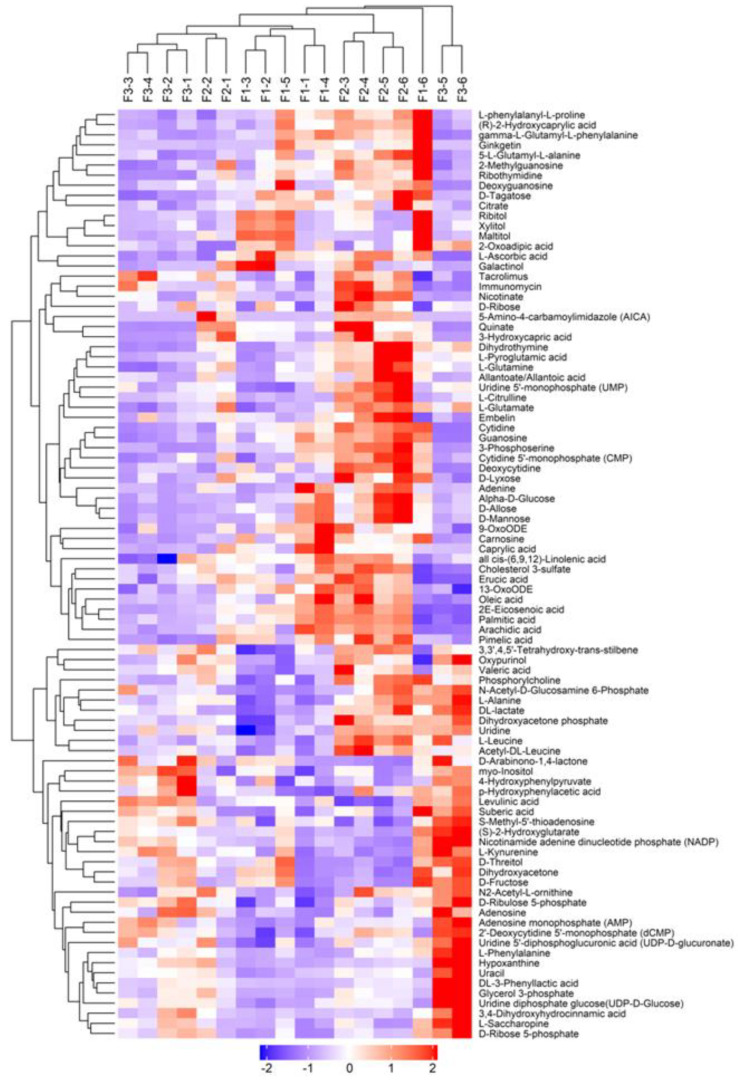
Hierarchical clustering of metabolite composition detected during the maturity stage of *A. cylindracea* fruiting body in the different treatments.

**Figure 6 jof-11-00132-f006:**
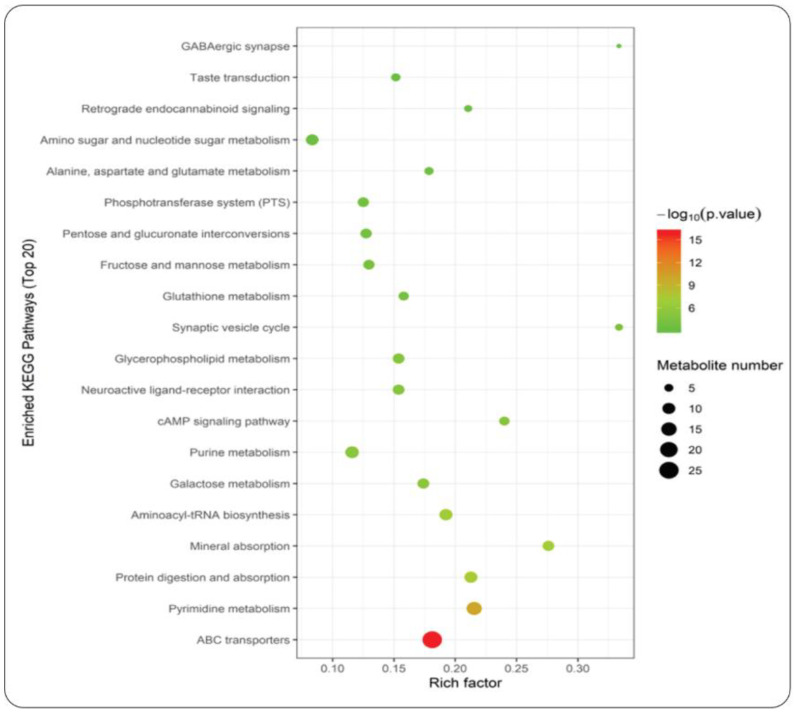
Bubble plot revealing the top 20 statistically significant KEGG pathways of metabolites in *A. cylindracea* grown in the WTL + SD vs. GJ + DD substrates.

**Figure 7 jof-11-00132-f007:**
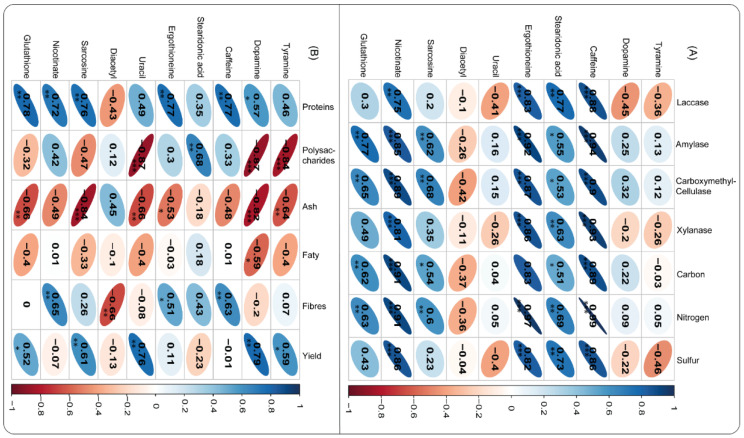
Pearson’s correlation coefficients reveal the associations among metabolites, physicochemical (**A**) biocomponents properties, and productivity (**B**) of *A. cylindracea* fruiting bodies are subjected to the various substrates. Statistical significance levels: * *p* < 0.05, ** *p* < 0.01, *** *p* < 0.001.

**Figure 8 jof-11-00132-f008:**
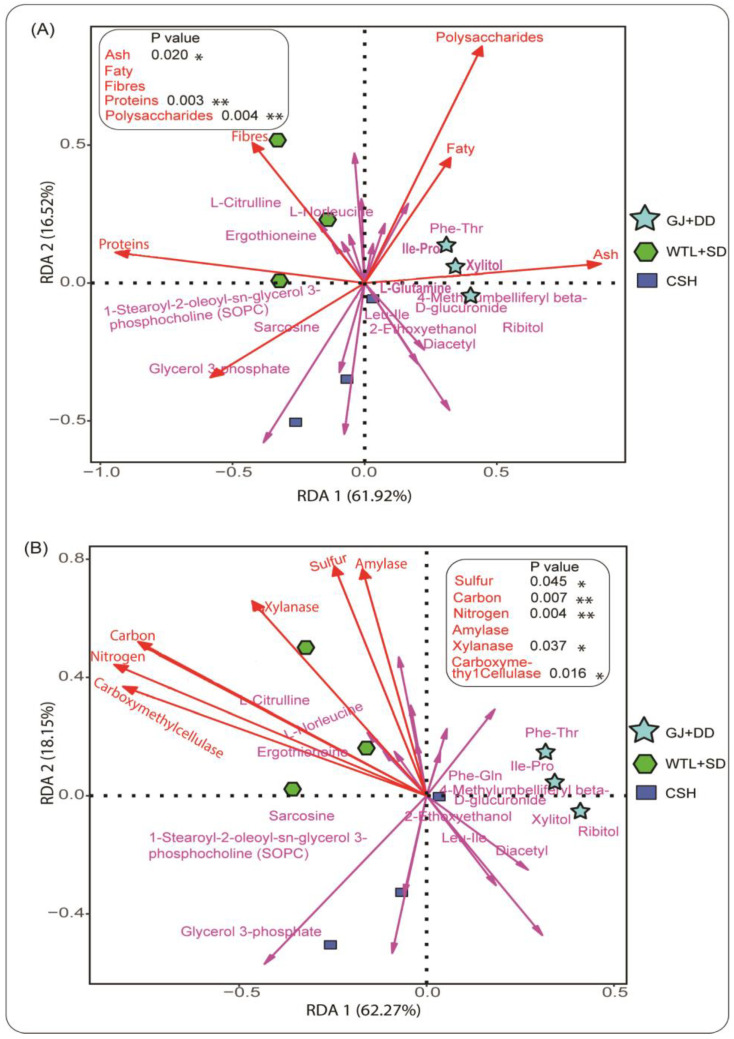
RDA illustrates the pattern and association between metabolites and biocomponents properties (**A**), followed by metabolites and physicochemical properties (**B**) detected in the *A. cylindracea* fruiting bodies. Statistical significance levels: * *p* < 0.05, ** *p* < 0.01.

**Table 1 jof-11-00132-t001:** Formulation of culture materials (%).

No.	Giant Grass + *Dicranopteris dichotoma* (GJ + DD)	Wasted Tea Leaf + Sawdust (WTL + SD)	Cotton Seed Husks (CSH)	Wheat Bran	Lime
1	48 + 30			20	2
2		48 + 30		20	2
3			78	20	2

## Data Availability

The original contributions presented in the study are included in the article/Appendix A, further inquiries can be directed to the corresponding authors.

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
