# Peer review of "The Influence of the Divergent Substrate on Physicochemical Properties and Metabolite Profiling of Agrocybe cylindracea Cultivation"

_jof, 2025, doi:10.3390/jof11020132_

Round 1

Reviewer 1 Report

Dear authors

I am here with submitting of the review manuscript ID: jof-3401313
The influence of the divergent substrate on physicochemical, biocomponent properties and biological efficiency of Agrocybe cylindracea cultivation

The data presented in manuscript are new and interesting additionally to make the manuscript more complete I suggest a brief addition to the introductory part.

A review of the literature on the cultivation of this mushroom from 2014 to 2024 was given, but nowhere was any data presented on the actual production of the species Agrocybe cylindracea. In this sense, I suggest the authors try to find data on the quantity obtained from cultivating this species at the country or world level.

The work represents new data and results that the various possibilities of application of plant as well as its components.

Dear authors

I am here with submitting of the review manuscript ID: jof-3401313
The influence of the divergent substrate on physicochemical, biocomponent properties and biological efficiency of Agrocybe cylindracea cultivation

The data presented in manuscript are new and interesting additionally to make the manuscript more complete I suggest a brief addition to the introductory part.

A review of the literature on the cultivation of this mushroom from 2014 to 2024 was given, but nowhere was any data presented on the actual production of the species Agrocybe cylindracea. In this sense, I suggest the authors try to find data on the quantity obtained from cultivating this species at the country or world level.

The work represents new data and results that the various possibilities of application of plant as well as its components.

Author Response

          Cover letter to reviewer’s comments

Thank you very much for taking the time to review this manuscript and providing useful comments. Kindly find the detailed responses below and resubmit the revised version.

Comments 1: A review of the literature on the cultivation of this mushroom from 2014 to 2024 was given, but nowhere was any data presented on the actual production of the species Agrocybe cylindracea. In this sense, I suggest the authors try to find data on the quantity obtained from cultivating this species at the country or world level.

Response 1: Thank you for pointing this comment out. We agree with the comments. The introduction section was revised according to the useful comments. The changes can be found on page 2 Paragraph one lines 48-55.

Thank you very much!!!!!

Reviewer 2 Report

The higher fungi, especially from Basidiomycota phylum, are the focus of many scientific researches because of their ability to sintesize a plethora of metabolites demonstrating a wide range of biological activities that could result in medicinal and nutraceutical application.

So from this point of view the research presented to me is actual but I have too many remarks that according me have to be taken in consideration in order this manuscript being accepted for publication.

1.     The title of the manuscript does not correspond to its content.

2.     The introduction is very poor, without any comprehensiveness. “Mushrooms are large fungi ….” !!!!!

3.     The aim of this research is not mention.

4.     Why the exact substrates were chosen to be used?

5.     Do the authors have information about the main components content of these substrates? Don't they think it was good to define the component composition?

6.     Let the authors explain what exactly do they mean under biological efficiency?

7.     The determination of the enzyme activity is bad and unclear described. For example „1g of the sample was measured, and 1 mL of the extract was added”.. Sample means what and what kind of extract. Most of the studied enzymes are extracellular and first the enzymes should be extracted in a suitable way and only then the enzyme activity should be determined using a suitable substrate.

8.     It is not clear how carbon dioxide emission is measured and why it is necessary?

9.     Nutritional composition: How the polysaccharides are determined?

10.  The results are not clear presented and the discussion is poor!

no

Author Response

Reply to the reviewer’s comments

Thank you very much for taking the time to review this manuscript and providing useful comments. Kindly find the detailed responses below and resubmit the revised version.

Comments 1:  The title of the manuscript does not correspond to its content.

Response 1: Thank you for pointing this comment out. We agree with the comments. The title of the manuscript has been revised according to the useful comments. The changes can be found in the revised manuscript title.

Comments 2: The introduction is feeble, without any comprehensiveness. “Mushrooms are large fungi ….”!!!!!

Response 2: Thank you for pointing this comment out. We agree with the comments. The introduction section was revised according to the useful comments. The changes can be found in the introduction section lines 38-91

Comments 3: The aim of this research is not mentioned.

Response 3: Thank you for pointing this comment out. The aim of this research was revised and improved. The changes can be found on page 2 Paragraph 3 lines 86-91.

Comments 4: Why the exact substrates were chosen to be used?

Response 4: Thank you for pointing this comment out. According to the previous research reports We have chosen these types of substrates because of their lignocellulosic composition, which A. cylindracea naturally breaks down in its habitat. They provide essential carbon sources, particularly lignin and cellulose, that support both mycelial growth and the development of fruiting bodies. Secondly, these substrate materials provide optimal physical properties for mushroom growth such as adequate water retention capacity, proper aeration through their natural porosity and suitable bulk density for mycelial colonization which is necessary to grow Agrocybe cylindracea mushrooms that we need to do the research on.

Comments 5: Do the authors have information about the main components content of these substrates? Don't they think it was good to define the component composition?

Response 5: Thank you for pointing this comment out. The researchers have reported about substrate compositions such as cellulose content, hemicellulose content, lignin content total carbon total nitrogen and C/N ratio which are very important in mushroom growth. In this study, we have analysed those compositions and the findings were demonstrated in the results section.

Comments 6: Let the authors explain what exactly they mean under biological efficiency.

Response 6: Thank you for pointing this comment out. Biological efficiency (BE) in mushroom cultivation is a measure of how effectively a mushroom strain utilizes a given substrate to produce the fruiting body. It's expressed as a percentage and is calculated by comparing the weight of the harvested mushrooms to the dry weight of the substrate used as mentioned in the data analysis part.

Comments 7: The determination of the enzyme activity is bad and unclear described. For example „1g of the sample was measured, and 1 mL of the extract was added”.. „Sample“ means what and what kind of „extract“. Most of the studied enzymes are extracellular and first, the enzymes should be extracted in a suitable way and only then the enzyme activity should be determined using a suitable substrate.

Response 7: Thank you for pointing this comment out. The section was well revised according to the useful comments. The changes can be found on page 4 Paragraph 1 lines 139-141.

Comments 8: It is not clear how carbon dioxide emission is measured and why it is necessary.

Response 8: Thank you for pointing this comment out. The methodology section was revised accordingly, and CO2 is necessary in mushroom cultivation due to the CO2 emission directly correlates with the metabolic activity of the mushroom. Knowing CO2 emission as a researcher helps monitor the progression of colonization and fruiting. In our research, the high CO2 levels typically mean more active growth, this helps us track different developmental stages (mycelial growth, pinning, fruiting) of agrocybe cyindracea grown in different substrates we have used.

Comments 9: Nutritional composition: How the polysaccharides are determined?

Response 9: Thank you for pointing this comment out.The methodology of the section of polysaccharide determination was well-revised according to the useful comments. The change can be found on page 5 Paragraph 1 lines 184-190.

Comments 10:   The results are not clearly presented, and the discussion is poor!

Response 10:  Thank you for pointing out this comment. The results and discussion sections were well revised. The change can be found in the resubmitted manuscript.

Thank you very much!!!

Round 2

Reviewer 2 Report

The authors take in considaration my remarks!

none

Author Response

          Cover letter to Academic editor’s comments

Thank you very much for taking the time to review this manuscript and providing useful comments. Kindly find the detailed responses below and resubmit the revised version.

Comment: There is still room for improvement in the data analysis section. Although the data analysis method is described in the “Materials and Methods” section, it is recommended that the specific data analysis method used be indicated in each annotation within the Figure legend. For example, the first figure does not meet the requirements for publication. When determining enzyme activity and substance content, standard deviation (SD) and significance analysis should be provided to enhance the reliability and persuasiveness of the data. Additionally, the resolution of this figure is low, and the pixel quality is insufficient, which affects the clarity and readability of the chart. The authors are advised to reprocess or replace it with a higher-resolution image to meet the publication standards.

Response: Figure one has been revised based on valuable comments, and the standard deviation (SD) has been added. The data analysis section has been thoroughly revised, the changes can be found on pages 238-239. The resolution and pixel quality of the figures have been enhanced. Consequently, the newly revised manuscript now includes 8 figures instead of 7, with updates made to figures 1, 2, and 3 in this resubmission.

Thank you very much!!!!!
